# LayerMix Law: Scaling Law for Large Language Models on Quality-Weighted Mixture Data with Repetition

## Abstract

Upweighting high-quality data in large language model (LLM) pretraining typically improves performance. However, the limited availability of high-quality data—particularly in overtrained regimes—means that stronger upweighting often increases repetition, which can degrade performance. This creates a fundamental trade-off between data quality and data repetition. In this paper, we systematically investigate how varying data quality and repetition affects models across different scales. Concretely, we partition the source corpus into buckets based on quality scores and sample from each bucket with different weights, thereby constructing training sets with diverse scales, quality distributions, and repetition levels. We then train a family of models on these datasets to measure performance across conditions. Building on these observations, we introduce a theoretical framework analogous to scaling laws, which we call **LayerMix Law**. LayerMix Law predicts model loss as a function of consumed tokens, model size, sampling weights, and repetition levels. The key intuition is to view training as the accumulation of information from data, where the amount of information is governed by data quality, while model scale and repetition determine the information gained per training step. We show that LayerMix Law accurately predicts the model performance on unseen data recipes at larger computation scale (up to 7B parameter run with 425B token, each x2 invest compute), with 0.15% average absolute error and 0.96% maximum absolute error, which enables efficient search for optimal data recipes without costly additional experiments. Moreover, LayerMix Law extrapolates reliably to different degrees of overtraining, providing a efficient tool for selecting data recipes under varying computational budgets.

## 1 Introduction

Training large language models (LLMs) requires access to high-quality data (Brown et al., 2020a; Chowdhery et al., 2023). However, the availability of high-quality data is severely limited (Villalobos et al., 2024), and in the data-constrained settings, upweighting higher-quality data inevitably increases repetition, which has been shown to impair performance when excessive (Muennighoff et al., 2023). This issue is further exacerbated by the widespread adoption of overtraining (Touvron et al., 2023; Yang et al., 2025)—a strategy that reduces inference costs compared to the compute-optimal regime (Hoffmann et al., 2022).

To address the shortage of high-quality data as model scale increases, a common compromise is to incorporate lower-quality data, thereby reducing the repetition of high-quality samples. Intuitively, high-quality data provides greater performance gains than low-quality data upon first exposure, but as repetition increases, the marginal benefit decays—eventually approaching that of unseen low-quality data. However, the optimal balance between quality and repetition remains unclear. A standard approach for identifying optimal mixing strategies is to run smaller-scale experiments and extrapolate performance to larger compute budgets using scaling laws (OpenAI et al., 2024; Hoffmann et al., 2022; Chowdhery et al., 2023). Yet, as shown in Figure 1, under conditions of data repetition, standard scaling laws fail to reliably predict model performance at scale (Hernandez et al., 2022; Muennighoff et al., 2023). Moreover, they do not generalize across different mixing

strategies , necessitating grid searches over data recipes—an approach that is costly even at small scales.

In this paper, we study the problem of scaling large language models in a data-aware regime, where training data consists of a heterogeneous mixture with varying quality levels, and each quality level is repeated to different extents. We introduce a theoretical framework, the LayerMix Law, which accounts for both the scaling effects of mixture weights and the impact of repetition. Our formulation views training as a process of accumulating information from the dataset, with model performance determined by the total information gained by the end of training. At each step, the information gain is modeled as the sum of contributions from different quality ranges. Within each quality range, the gain depends on two factors: an information density function, parameterized by quality (with higher quality assigned higher density), and an exponential decay term that captures the interactions between model scale, data scale, and repetition level.

To fit the parameters of the LayerMix Law, we construct a suite of datasets that vary along three axes: scale, quality, and repetition level. Specifically, we partition the source dataset into buckets according to quality scores, and then sample from each bucket with different weights, a procedure we refer to as LayerMix sampling. Following the data-constrained setting, the source dataset is first downsampled to the target scale to ensure stable repetition effects. We then train 9 models ranging from 252M to 1.2B parameters from scratch, each under the same 3.6x over-trained ratio (Gadre et al., 2024). For each model, we construct three datasets with distinct LayerMix sampling configurations, resulting in 27 total training runs. Model performance is evaluated as the average perplexity across five downstream tasks. Finally, we fit the LayerMix Law to these results, estimating the parameters that best capture the relationship between information gain and observed performance.

To verify the generalization ability of LayerMix Law, we conduct three extrapolate experiments. Firstly we create datasets with two other unseen LayerMix sampling parameters to test the extrapolation on unseen data recipe. Secondly, we conduct experiments with larger computation scale on both seen and unseen LayerMix sampling parameters to test the scaling ability. Thirdly, we conduct experiments with larger over-trained ratio (25x), to test the generalization ability of LayerMix Law parameters. We found that LayerMix Law is able to predict loss on unseen data recipes at different scales (up to 7B parameter run with 425B token, each x2 invest compute), with 0.15% average absolute error and 0.96% maximum absolute error. We search for the optimal data recipe on 2.5B model with LayerMix Law with no additional experiments, and the result achieves the best among 4 other random ablations. Also on over-trained ratio, directly apply LayerMix Law yields good scaling results.

## 2 RELATED WORK

**Scaling Laws**   Scaling of transformer language models (Vaswani et al., 2017) and training data has been shown to provide consistent performance improvements (Chowdhery et al., 2023; Radford et al., 2019). This has led to the development of many large-scale models, including both dense architectures (Brown et al., 2020b; Rae et al., 2021; Grattafiori et al., 2024) and mixture-of-experts (MoE) variants (DeepSeek-AI et al., 2025; Yang et al., 2025; Fedus et al., 2021). Early empirical studies observed that neural networks exhibit predictable power-law scaling behavior (Hestness et al., 2017). Building on this, Hoffmann et al. (2022) investigated the compute-optimal setting, suggesting that model size and training data should be scaled in roughly equal proportions, whereas Kaplan et al. (2020) proposed a different allocation strategy emphasizing alternative trade-offs. More recent work, such as DeepSeek-AI et al. (2024), further explored how compute budget $C$ interacts with optimization hyperparameters, including the choice of batch size and learning rate.

Recently, there has been a growing trend of over-training smaller models on large datasets (Touvron et al., 2023; Yang et al., 2025), motivated by both efficiency considerations and deployment constraints. Sardana et al. (2024) extended the Chinchilla framework by incorporating data quality and inference requirements, deriving optimal allocations between model size and dataset scale under these additional factors. Complementary to this, Gadre et al. (2024) demonstrated that scaling laws remain reliable even in the over-trained regime, where models are trained significantly beyond the compute-optimal point. These findings highlight the importance of revisiting scaling strategies in regimes constrained by data availability, quality variation, or inference efficiency demands.

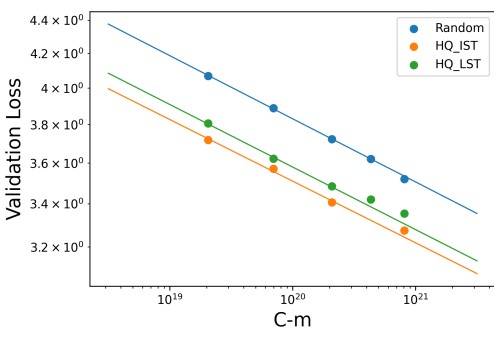 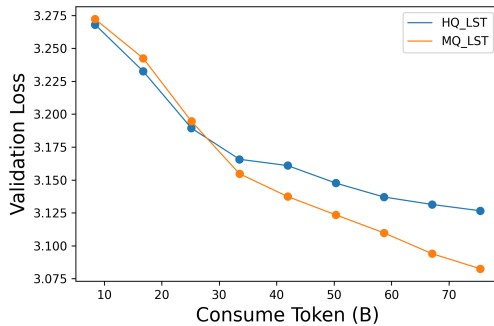

(a) $Loss$-$C_m$ Curve of model trained with random data and Layermix packed data

(b) The validation loss of different LayerMix experiments in training process.

Figure 1: Effects of quality selection and data repetition

For predicting downstream performance, Isik et al. (2025) demonstrated that downstream metrics also exhibit predictable scaling effects after fine-tuning, extending the scope of scaling laws beyond pre-training loss. Schaeffer et al. (2023) further established a connection between non-linear evaluation metrics and model perplexity, providing a more stable predictor of performance compared to prior approaches such as Wei et al. (2022), which observed instability in emergent metrics. Ge et al. (2025) employed fine-grained alignment between a model's foundational capabilities and the requirements of specific downstream tasks, leading to more accurate scaling law predictions for that task. Ruan et al. (2024) leveraged performance data from existing models to predict complex behaviors and emergent abilities by modeling them in a shared, low-dimensional capability space. Both approaches highlight a growing trend toward understanding scaling phenomena through the lens of modular capabilities.

**Data-Aware Scaling**    Traditional scaling laws typically assume that training data is both fixed and unlimited. In practice, however, high-quality data is scarce and often upsampled to improve model performance (Lin et al., 2022). Xue et al. (2023) found that continuing to train on repeated data is generally preferable to stopping early. This aligns with earlier findings by Hernandez et al. (2022) and Muennighoff et al. (2023), who also observed performance deterioration when upsampling or repeating datasets. These insights underscore the limitations of classical scaling laws in data-constrained settings and highlight the need for more sophisticated, data-aware approaches. More recently, Chen et al. (2025) studied how sub-scaling laws interact with data density, providing a finer-grained understanding of scaling under limited-data regimes.

Other lines of work have applied scaling laws to guide the design of optimal data recipes. For instance, Ye et al. (2025) incorporated data mixture weights into the functional form of loss prediction, while Kang et al. (2025) suggested that optimal mixing strategies themselves depend on model scale. Liu et al. (2024) employed a proxy model to predict the final model's performance under different mixture ratios, thereby discovering high-quality data recipes without training large-scale models. Gu et al. (2024); Que et al. (2024) investigated scaling laws in the context of continued pre-training, using them to inform domain mixture strategies. In addition, Chang et al. (2024) analyzed how scaling interacts with data quality. In contrast to these approaches, our work aims to predict model loss on mixtures of varying quality and repetition levels, thereby providing a more general framework for data-aware scaling.

## 3    LIMITATIONS OF CONVENTIONAL SCALING LAWS

In this section, we reveal and substantiate a critical limitation of conventional scaling laws in the context of data repetition and quality selection. First, we introduce the LayerMix sampling function in section 3.1, to imitate real scenario where the data is a mixture of different quality and repetition degrees. Next, we compare the relationship between the model's loss $L$ and amount of compute $C$ in cases with and without repetition in section 3.2, and the results show that the traditional scaling law performs well on data without repetition

Table 1: Preset LayerMix sampling weights and Searched optimal sampling weights for 2.5B model.

| Name | $w1$ | $w2$ | $w3$ | $w4$ | $w5$ | $w6$ |
|---|---|---|---|---|---|---|
| **HQ (High Quality)** | 0.80 | 0.10 | 0.03 | 0.03 | 0.02 | 0.0 |
| **MHQ (Medium-High Quality)** | 0.66 | 0.22 | 0.05 | 0.03 | 0.02 | 0.0 |
| **MQ (Medium Quality)** | 0.48 | 0.23 | 0.13 | 0.07 | 0.07 | 0.0 |
| **MLQ (Medium-Low Quality)** | 0.38 | 0.21 | 0.20 | 0.11 | 0.08 | 0.0 |
| **LQ (Low Quality)** | 0.24 | 0.20 | 0.19 | 0.18 | 0.17 | 0.0 |
| **Optimal Recipe of 2.5B model with** $m = 3.6$ | 0.50 | 0.49 | 0.01 | 0.0 | 0.0 | 0.0 |

## 3.1 LAYERMIX SAMPLING FUNCTION

**Source Data**  We obtain our training corpora from Common Crawl (Common Crawl Foundation), following similar filtering process as Penedo et al. (2023) and obtain 15T English tokens. We ran global fuzzy deduplication across all snapshots to ensure there is no repeat data in the corpora. The final dataset contains 3.7T token. The details are in Appendix A.

**Training Data Sampling**  We first define the quality score for each document following Liu et al. (2025), where two quality classifiers (Penedo et al., 2024; Li et al., 2025) are applied and the final quality score is averaging the normalized score from each classifier. With the quality score, We then rank all documents based on quality score and empirically split the training corpora into six quality buckets based on their quality rankings, which are 0-5%, 5%-20%, 20%-40%, 40%-60%, 60%-80%, 80%-100%.

Then we define a LayerMix sampling function $H(w, K, S, B)$, where $S$ represents the packing source tokens and $K$ represents the packed tokens for training data, and we perform 1 epoch training to avoid additional repetition factors. $w = [w_0, w_1, ..., w_5]$ and $\sum w_d = 1$, representing the token proportion of the data in the $d$-th bucket in the packed training data, and $B = [B_0, B_1, ..., B_5]$ is the token proportion of source token for each data bucket, in our experiments, $B = [5\%, 15\%, 20\%, 20\%, 20\%, 20\%]$. The LayerMix sampling function $H$ returns a dataset of total token $K$ with bucket proportions given by $w$ sampled from a corpora of total token $S$. In the returned training data, the $d$-th bucket have $w_d K$ tokens, which are sampled from $B_d S$ tokens in source data, leading to $M_d = min(w_d K, B_d S)$ unrepeated tokens. Then the average repeat time for data in the $d$-th bucket is $R_d = \frac{M_d}{B_d S}$. The detailed algorithm is shown in Appendix B.

With different setting of $w, K, S$, the LayerMix sampling function generate datasets with different scales, quality and degrees of repetition. We ensure $w_d$ is greater than $w_{d+1}$ to guarantee higher quality buckets have a larger proportion in the training data. We select 5 settings of $w$, representing different levels of quality and repetition, namely HQ (High Quality), MHQ(Medium-High Quality), MQ (Medium Quality), MLQ (Medium-Low Quality), LQ (Low Quality). The detailed proportion is as Table 1. $w_5$ is set to 0 so that we drop the bottom 20% of data ranked by quality score. Throughout work, we set $K = S$ to reduce complexity unless mentioned, so that we only focus on the repetition change caused by $w$.

## 3.2 TRADITIONAL SCALING LAW BETWEEN LOSS AND AMOUNT OF COMPUTE

We compare the relationship between model loss $L$ and total compute $C$ under regimes with and without repetition in an overtrained setting. Specifically, under the compute-optimal scheme, $C_{opt} = N_{opt} K_{opt}$, where $K$ is the consume token, $N$ is the non-embedding FLOPs per token as defined in DeepSeek-AI et al. (2024) and $N_{opt}, K_{opt}$ is the Chinchilla-optimal pair. Then in the overtrained setting, following Gadre et al. (2024), we set $K_m = \sqrt{m} K_{opt}$, $N_m = \frac{1}{\sqrt{m}} N_{opt}$, $C_m = K_m N_m$ with $m = 3.6$. And Gadre et al. (2024) shows that the the Loss–Compute relation preserves the fitted exponent for models trained with the same overtraining factor $m$.

**Random**  The training data is randomly sampled from sufficient large source data, where $S \gg K$, meaning there is rarely no repetition in the training data.

**HQ_IST** We use the layermix sampling function and $w$ is set to mainly focus on high quality data, namely HQ (High Quality), see details in Appendix B. Then we set $S \gg K$, denoting as IST (Infinite Source Token), meaning there is almost no repetition in train data

**HQ_LST** We use the layermix sampling function and set $S = K$, denoting as LST (Limited Source Token), where there exists repetition in the training data.

We plot the log-log plot between $C_m$ and model loss $L$ in Figure 1a. Note that the loss $L$ mentioned here, including later references, is the average loss of the model on the following five downstream tasks: HellaSwag (Zellers et al., 2019), ARC-E, ARC-C (Clark et al., 2018), MMLU (Hendrycks et al., 2021), TriviaQA (Joshi et al., 2017). Following Schaeffer et al. (2023), we transfer the accuracy on downstream tasks into perplexity for better scaling effect. The results show that for random dataset, the scaling law loss-C curve can fit all of the data points. But for HQ_IST and HQ_LST, the performance decay as compute $C_m$ scales. More results are detailed in Appendix F

The above observation indicating that traditional scaling law only holds for completely random and non-repeated data. The change of data quality distribution or repeat time undermines its predictive accuracy. So we need a modified scaling law that incorporates both data quality distrubution and the degree of data repetition as core variables.

## 4 LAYERMIX LAW

Scaling laws for large language models traditionally rely on compute, but often fail to account for effects of data quality and repetition. In this section, we introduce the design of LayerMix Law. We treat the training process as gaining information from the dataset and propose to calculate Information Quantity as accumulation of information gain throughout the training process, which synthesizes the impacts of data quality, repetition level, model scales and total training tokens, and then build power-law relationship with the model's final validation loss.

### 4.1 INFORMATION QUANTITY

We first show how the evaluation loss changes during training of two 850M models trained on two datasets packed with different LayerMix sampling weights in Figure 1b. The dataset HQ_LST repeates top 5% quality data for about 16 times and MQ_LST for 10 times, MQ_LST has lower quality than HQ_LST but less repetition. By default, we use the LST setting and we ignore the notation of LST for simplicity unless mentioned. In the early training stage, HQ and MQ experiments achieves almost same evaluation loss. However, later in the training, the loss of HQ experiment decreases much more slowly, resulting in a worse final performance compared to the MQ experiment, indicating that more repetition expedites the decay of performance gaining.

Based on this observation, we propose an exponential decay function to model the decreasing information gain of repeated data. Assuming the Information Quantity a document $i$ contains is $I_i$, then the information a language model gets at $t$-th learning from the document $i$ is:

$$I_{i\_\text{part}}(t, \lambda_N) = I_i \cdot \lambda_N e^{-\lambda_N t} \tag{1}$$

where $\lambda$ is a hyperparameter which is related to the non-embedding FLOPs/token $N$.

When a langue model learning the document for total $T$ times, the Information Quantity learned from the document is:

$$I_{i\_\text{total}}(T, \lambda_N) = \int_0^T I_{i\_\text{part}}(t, \lambda_N)dt = I_i \cdot (1 - e^{-\lambda_N T}) \tag{2}$$

Equation 2 captures the principle of diminishing returns in learning: repeated exposure to a document yields progressively smaller gains, causing the total acquired information to saturate and asymptotically approach the document's full information content $I_i$.

Considering in large-scale training, due to the forgetting effect: as the total train token $K$ grows, the average Information Quantity language model acqiure from a singel sample decreases. We introduce train token $K$ to the Equation 1 to accommodate this phenomenon:

$$I_{i\_\text{part}}(t, \lambda_N, K) = I_i \cdot \lambda_N e^{-\lambda_N t / \log(K)} \tag{3}$$

Then the Equation 2 becomes to:

$$I_{i\_\text{total}}(t, \lambda_N, K) = \int_0^T I_{i\_\text{part}}(t, \lambda_N, K)dt = I_i \cdot \log(K)(1 - e^{-\lambda_N T / \log(K)}) \tag{4}$$

For all the training data, we sum them together as the final Information Quantity the language model learned from the training corpora, denoting as $info$:

$$info(w, K, S, f, \lambda_N) = \sum_d I_d \cdot \log(K)(1 - e^{-\lambda_N R_d / \log(K)})$$

$$= \sum_d f_d M_d \log(K) \cdot (1 - e^{-\lambda_N R_d / \log(K)}) \tag{5}$$

where $d$ is the quality bucket number from 0 to 5. $I_d$ is the total information quantity in d-the bucket, which can be calculated by the multiplication of number of unique tokens $M_d = min(w_d K, B_d S)$ and information density $f_d$, which is a parameterized quality density function. $R_d = \frac{w_d K}{M_d}$ is the average repeat times for the data from the $d$-th bucket and $\lambda_N$ is related with $N$, which are to be fitted from the data.

Equation 5 can be divided into two parts: the first term is $I_d = f_d M_d \log(K)$, it represents the total Information Quantity contained in the packed data of the $d$-th bucket, and the second term is $1 - e^{-\lambda_N R_d / \log(K)}$, it represents the language model's learning ability on this data when repeated an average of $R_d$ times. And the total Information Quantity learned by the language model is the product of these two terms.

We propose Information Quantity, a metric computed from LayerMix sampling weights $w$, train token $K$ and two fitted functions ($f_d$, $\lambda_N$, to quantify the knowledge learned during training. Since it is designed to be monotonic with model performance, it enables loss prediction for various training configurations prior to any actual runs. The fitting of $f_d$ and $\lambda_N$ is described in Section 5.2.

## 4.2 INFORMATION-LOSS SCALING LAW ON REPEATED DATA

As illustrated in Figure 1a, the evaluation loss of the model trained with repeated data is higher than that of the model trained without repeated data, and the traditional scaling law fails to predict language model's performance under this circumstance.

We use the Information Quantity proposed in Section 4.1 and plot the $L$-$info$ figure. As illustrated in Figure 2, when we replace the traditional computation axis $C$ with our novel metric: Information Quantity, the experimental points with different LayerMix sampling weights $w$, Model non-embedding FLOPs/token $N$ and Train Token $K$ now collapse perfectly onto a single, unified power-law curve, where they were previously scattered and separated.

Then the relationship between the loss $L$ and $info$ can be measured using power-law formulation as:

$$L = \alpha \cdot info^{-\beta} \tag{6}$$

In our experiment, $\alpha = 3.7373$ and $\beta = 0.0441$. We show them in a log-log plot, so it appears as a straight line with a slope of $-\beta$ and an intercept of $\log(\alpha)$.

Like the traditional scaling law (Hoffmann et al., 2022), we can now conduct experiments on small models to compare the advantages and disadvantages of different experimental configurations, and

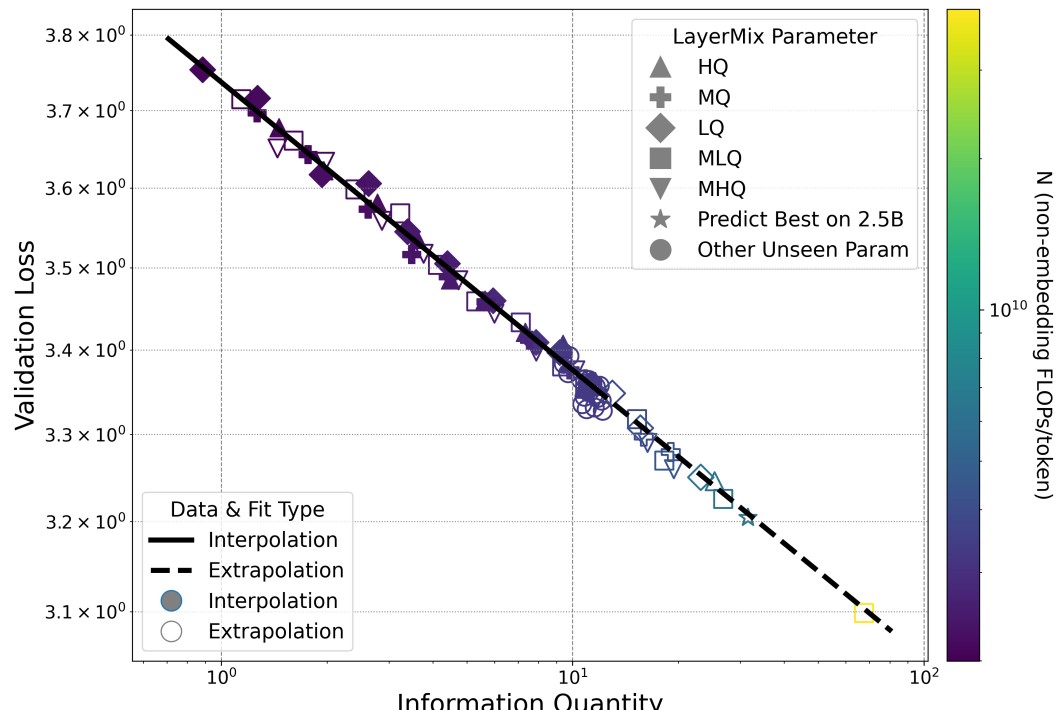

Figure 2: The Unified Information-Loss Scaling Law. The scaling law is fitted only on the interpolation data (solid markers), yet it accurately predicts the performance of held-out extrapolation data (hollow markers).

then use our proposed information scaling law to extrapolate the performance of larger models under larger training tokens.

## 5 FITTING EXPERIMENTS

### 5.1 TRAINING SETUP

We train 9 models ranging from 252M to 1.2B on 3 layermix sampling weights (*HQ, MQ,* and *LQ*), with 3.6x over-trained ratio, resulting in 27 experiment runs in total to collect data for fitting the LayerMix Law parameters. We use transformer architecture (Vaswani et al., 2017), SwiGLU (Shazeer, 2020) as the activation function and RoPE embeddings (Su et al., 2024). We use a tokenizer with 250k vocabulary. See Appendix B and Appendix C for details about LayerMix sampling weights, model structure, learnign rate and optimizer.

### 5.2 FITTING THE CURVE

In this section, we introduce how to fit the parameters in LayerMix Law to predict the model performance collected in Section 5.1. Since Information Quantity $info$ indicates the knowledge learned by the model, we expect larger $info$ to correspond to lower evaluation loss $L$. Considering that there may exist scale difference between $info$ and model loss $L$, we choose Spearman correlation $\rho_s$ as the fitting metric, i.e., the object is to find the optimal quality density $f$ and $\lambda_N$ such that the Spearman correlation between evaluation loss $L$ and $info$ is minimized for all the experiments over $N, w$:

$$(f^*, \lambda^*) = \operatorname*{argmin}_{f, \lambda} \sum^N \sum^w \left( \rho_s \left( L_N, info \left( w, K_N, S_N, f, \lambda_N \right) \right) \right) \quad (7)$$

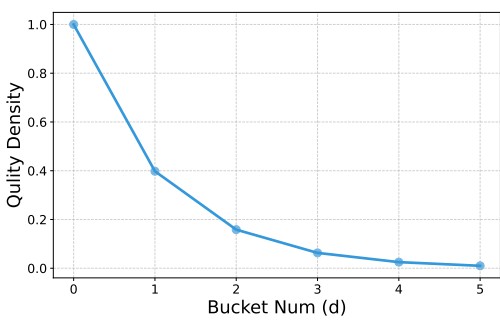
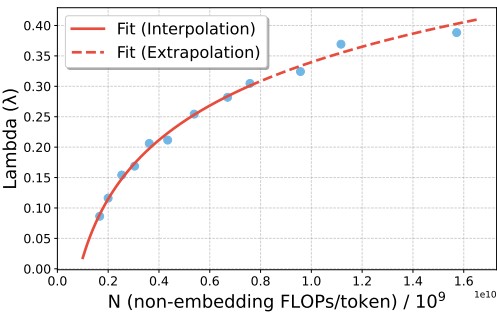

(a) The fitted quality density function $f_d$. The quality density is a monotonically decreasing function of the bucket index, meaning buckets with higher-quality data are assigned a higher density value.

(b) The relationship between $\lambda$ and $N$ with a fitted curve. The blue scattered points represent the observed data. The solid red line shows the fit within the data range, while the dashed line represents the extrapolation.

Figure 3: The fitted function of quality density function and relationship between $\lambda_N$ and $N$

To prevent from over-fitting, we make some assumption based on naive intuition. For $f$, as it indicates the quality density, the higher-quality bucket should have larger $f$. As smaller $d$ corresponds to higher-quality buckets, we define $f$ in the following form to ensure it is a decreasing function:

$$f_d(\theta) = e^{-\theta * d} \tag{8}$$

where $\theta$ is a hyperparameter and $\theta > 0$.

$\lambda_N$ is related to the model's learning capacity, so $\lambda_N$ should increase as $N$ increases. But we need to find the formula for $\lambda_N$ related with $N$ so that it can scale to larger $N$. To do this we first sample 100,000 combinations of $\theta$ and $\lambda_N$ from the parameter space, then select optimal $\theta^*$ and $\lambda_N^*$ based on Equation 7. The fitted quality density $f(\theta^*)$ is shown in Figure 3a with fitted $\theta^* = -0.922$.

Having the $\lambda_N^*$ values of different models, as is shown in Figure 3b, we try to fit the $\lambda_N$-$N$ curve. The relationship between $\lambda_N$ and $N$ is observed to be non-linear, exhibiting rapid growth for smaller $N$ and gradually saturating as $N$ increases. This trend is well-approximated by a logarithmic function. Therefore, we choose the $\lambda_N$-$N$ curve using following formula:

$$\lambda_N(a, b) = a \cdot \ln(N) + b \tag{9}$$

Using existing $\lambda_N^*$, we fit the $\lambda_N$-$N$ curve in Figure 3b with fitted $a^* = 0.140$, $b^* = 0.018$. To validate this fit, we compute $\lambda_N^*$ for larger $N$ under the fixed $\theta^*$, and examine whether these values lie on the predicted $\lambda_N$-$N$ curve. As illustrated in Figure 3b, the results demonstrate strong extrapolation performance, supporting the correctness of our formulation. We compared with different formats of 9 in Appendix E and the log function best fit the trend and extrapolates

Finally, with $f(\theta^*)$ and $\lambda_N(a^*, b^*)$, we can calculate the Information Quantity for arbitrary layermix sampling weights $w$, train token $K$, source token $S$ and model non-embedding FLOPs/token $N$.

## 6 EXTRAPOLATION

The LayerMix Law achieves a strong fit on the training data after parameter optimization, and we subsequently employ it to predict loss under the unseen conditions to assess its robustness. To rigorously evaluate its capability, we first compare it with the traditional scaling laws. Then we test the extrapolation along three key axes: (i) novel LayerMix sampling weights, (ii) larger computational scales, and (iii) varying degrees of over-training. Finally, we use our LayerMix Law to predict optimal data recipe under different training settings and valid the recipe by comparing with preset recipes.

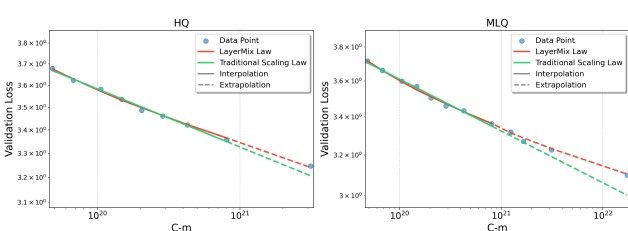 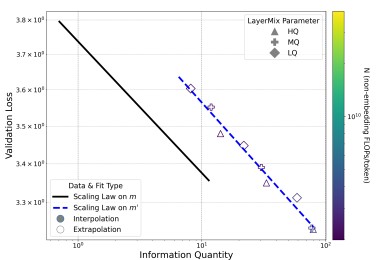

(a) Validation loss versus compute C in the loss–C view. Curves are fit on 252M–1.2B and extrapolated to larger models.

(b) Cross-Regime Prediction of the Scaling Law.

Figure 4: The extrapolation results of LayerMix Law

### Comparing with traditional scaling laws

Figure 4a contrasts our LayerMix law with the traditional power scaling law in the loss–$C$ plane. Both curves are fit using models in the 252M–1.2B range and then extrapolated to larger models. The Info curve tracks the MLQ data more closely within the fitting regime and remains accurate when extrapolating up to 7B models, avoiding the overly optimistic loss reductions predicted by the traditional law at high compute. Concretely, the traditional scaling law tends to under-estimate loss as $C_m$ grows, whereas the Info curve preserves a data-aligned decay that better matches the realized validation losses of larger models.

**Extrapolation to other LayerMix Sampling Weights**

We first test the ability to generalize to an unseen LayerMix sampling weights. We test on unseen dataset generated with $MLQ$, $MHQ$ on model scales ranging from $252M$ to $1.2B$, which are within the range of training data. Also we random sample 25 more sampling weights and run experiments on $1.2B$ model only.

The result is shown in Figure 2 marked by hollow squares (for $MLQ$) and hollow downward-pointing triangles (for $MHQ$). As can be seen, these points align remarkably well with the scaling law curve established by the initial $HQ$, $MQ$, $LQ$ data, demonstrating the predictive power of our model on unseen LayerMix sampling weights. The traditional scaling laws requires additional experiments on different data recipes to fit new curves, while ours can directly predict loss on unseen recipes.

**Extrapolation to Larger Models**

To test the extrapolation ability on model scale, we use the same Layermix sampling weights $MQ$,$LQ$ to train models ranging from 1.5B to 2.5B and $HQ$, $LQ$ to train model with 2.5B parameters, which are out of the range of training data. The experimental results of larger models are shown in Figure 2, we can see LayerMix Law predict the loss on larger scale accurately for all three sampling weights, proving the ability of scaling on model size.

**Combination of Extrapolation**

Further more, we combine the two extrapolation above and test the effectiveness on both unseen LayerMix sampling weights and unseen scales. We run experiments with $MLQ$,$MHQ$ on models ranging from $1.5B$ to $7B$. As shown in Figure 2, LayerMix Law also generalise well on these combined extrapolation condition. On all the unseen data points, including unseen LayerMix sampling weights and model scales, LayerMix Law predict the validation loss with $0.15\%$ average absolute error and maximum error is $0.96\%$. This proves that our proposed information scaling law has reliable extrapolation capability.

**Extrapolation to Larger Overtrain Degree**

To explore the model's reliability under varying sub-optimality, we conducted a second series of experiments at a higher overtrain degree, $m' = 25$. This new regime was anchored by a 1.2B model trained on 640B tokens (the $C_{m'}$ experiment), contrasting with our initial $C_m$ experiment anchored at 106B tokens.

Table 2: Results of four distinct LayerMix experiments on 2.5B model.

| Name | HQ | LQ | MLQ | Pred Best |
|---|---|---|---|---|
| Validation Loss | 3.246 | 3.250 | 3.226 | **3.204** |

Table 3: The best data recipe for different models and train token

| Model | Train Token | Source Token | $w1$ | $w2$ | $w3$ | $w4$ | $w5$ | $w6$ |
|---|---|---|---|---|---|---|---|---|
| | 500B | 500B | 0.496 | 0.492 | 0.007 | 0.003 | 0.002 | 0.000 |
| 7B | 800B | 500B | 0.439 | 0.430 | 0.130 | 0.001 | 0.000 | 0.000 |
| | 1000B | 500B | 0.395 | 0.387 | 0.214 | 0.003 | 0.001 | 0.000 |
| | 500B | 500B | 0.548 | 0.444 | 0.004 | 0.003 | 0.002 | 0.000 |
| 1.8B | 800B | 500B | 0.496 | 0.492 | 0.007 | 0.003 | 0.002 | 0.000 |
| | 1000B | 500B | 0.491 | 0.487 | 0.017 | 0.005 | 0.000 | 0.000 |
| | 500B | 500B | 0.619 | 0.376 | 0.004 | 0.001 | 0.000 | 0.000 |
| 1.2B | 800B | 500B | 0.496 | 0.492 | 0.007 | 0.003 | 0.002 | 0.000 |
| | 1000B | 500B | 0.496 | 0.492 | 0.007 | 0.003 | 0.002 | 0.000 |

For the $C_{m'}$-experiment, we calculated the Information Quantity using the same quality density $f(\theta^*)$ and $\lambda_N(a^*, b^*)$ fitted previously on the $C_m$ data. As shown in Figure 4b, the new experimental points align with a new scaling law curve. The resulting curves for $C_m$ and $C_{m'}$ appear nearly parallel, suggesting the overtrain degree $m$ primarily shifts the curve's intercept. This confirms that our proposed Information Scaling Law is effective across different overtrain degrees.

**Optimizing Data Recipe with LayerMix Law**

The ability of predicting loss on unseen data recipes and scales enables us to search for best data recipe without additional experiments. Similar to Liu et al. (2024). We randomly sample 100k LayerMix parameters from the parameter space, compute the information for each set of parameters, and convert it to loss via Equation 6. We then select the parameter that minimizes the predicted validation loss as the optimal LayerMix configuration for each training setting.

To verify the optimal recipe, we conduct experiments on $2.5B$ model with optimal data recipe and 3 other layermix sampling weights. The result optimal recipe is as in Table 1. As shown in Table 2, our optimal recipe achieves the best validation loss.

In Table 3, we present the optimal LayerMix parameters for different model sizes and training-token counts under a fixed source-token budget of 500B tokens. The optimal LayerMix parameters exhibit two clear trends. First, at a fixed training-token count, smaller models favor a higher fraction of high-quality data, whereas larger models benefit more from diversity and thus allocate a smaller fraction to the high-quality data. Second, as the total training tokens increase, the optimal LayerMix parameters shift from a high-quality emphasis toward greater diversity. More results are shown in Appendix H. In short: Small models or small training budgets prioritize quality; large models or large training budgets prioritize diversity.

## 7 CONCLUSION

In this paper, we propose a refined scaling law modeling **LayerMix Law**, which focus on predicting model performance on downstream tasks under data-constrained settings with weighted-quality mixing. The LayerMix Law provides accurate predictions of model performance on unseen data recipes at larger computational scales, achieving an average absolute error of only 0.15% and a maximum error of 0.96%. This enables efficient discovery of optimal data recipes without the need for extensive additional experiments. Furthermore, the LayerMix Law extrapolates reliably across varying degrees of over-training, offering an effective tool for selecting data recipes under different computational budgets.

## 8 ETHICS STATEMENT

Our research is based on the publicly available Common Crawl dataset. We do not foresee any direct negative societal impacts stemming from our methodology or the resulting models.

## 9 REPRODUCIBILITY STATEMENT

Our experiments are based on the open-source Common Crawl dataset. All experimental settings, model architectures, hyperparameters, and implementation details have been thoroughly described in the main body and the appendix to ensure that other researchers can independently reproduce our results based on this information.

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

# A TRAINING DATASET

We use the English portion of the Common Crawl Dataset (Common Crawl Foundation), utilizing 96 of the snapshots, from CC-MAIN-2013-20 to CC-MAIN-2024-18. Following Bi et al. (2024), we ran a global fuzzy deduplication across all snapshots, resulting in a total dataset with 3.7T tokens.

# B LAYERMIX SAMPLING FUNCTION

We show the detail of LayerMix sampling function in Algorithm 1.

---

**Algorithm 1** LayerMix Sampling Function $H(w, K, S, B)$

---

1: **function** H$(w, K, S, B)$
**Require:**
  $w$: A list of target proportions for six buckets, $w = [w_0, ..., w_5]$, where $\sum w_d = 1$.
  $K$: The total number of tokens for the final training dataset.
  $S$: The total number of tokens in the entire source corpora.
  $B$: The source distribution proportions B=[0.05, 0.15, 0.2, 0.2, 0.2, 0.2].
**Ensure:**
  $D_{train}$: The final packed training dataset.
  $M$: A list of unique token counts for each layer, $M = [M_0, ..., M_5]$.
  $R$: A list of average repetition counts for each layer, $R = [R_0, ..., R_5]$.

2:  Initialize an empty training dataset $D_{train} \leftarrow \emptyset$.
3:  Initialize empty lists for statistics: $M \leftarrow [], R \leftarrow []$.

4:  **for** $d \leftarrow 0$ **to** 5 **do**               ▷ Iterate through each quality bucket
5:    $K_{needed} \leftarrow K \times w_d$   ▷ Calculate tokens needed from bucket $d$ for the target mix
6:    $S_d \leftarrow S \times B[d]$           ▷ Calculate source tokens available in bucket $d$
                       ▷ Calculate the sampling ratio for the current bucket
7:    $Ratio_d \leftarrow K_{needed}/S_d$
                       ▷ — Detailed sampling process for bucket $d$ —
8:    Initialize an empty temporary set $D_{sampled\_d} \leftarrow \emptyset$.
9:    **for all** data point $x$ in bucket $d$ **do**
                 ▷ 1. Deterministic copy for the integer part of the ratio
10:      **for** $i \leftarrow 1$ **to** $\lfloor Ratio_d \rfloor$ **do**
11:        Add $x$ to $D_{sampled\_d}$
12:      **end for**
                  ▷ 2. Probabilistic sampling for the fractional part
13:      **if** $Ratio_d - \lfloor Ratio_d \rfloor > 0$ and $random() < (Ratio_d - \lfloor Ratio_d \rfloor)$ **then**
14:        Add $x$ to $D_{sampled\_d}$
15:      **end if**
16:    **end for**
17:    Append all data from $D_{sampled\_d}$ to $D_{train}$.

18:    $M_d \leftarrow \min(K_{needed}, S_d)$     ▷ Calculate unique tokens based on the new formula
19:    Append $M_d$ to $M$.
20:    $R_d \leftarrow K_{needed}/M_d$            ▷ Calculate average repetition count
21:    Append $R_d$ to $R$.
22:  **end for**

23:  **return** $D_{train}, M, R$                ▷ Return dataset and statistics
24: **end function**

---

$$\sqrt{m} = \frac{N_{opt}}{N} = \frac{D}{D_{opt}} \tag{10}$$

A value of $m = 1$ indicates a compute-optimal training run, while $m > 1$ signifies that the model is overtrained relative to its compute budget.

---

**Algorithm 2** Calculation of Overtrain Degree and Optimal Tokens

---

1: **function** CALCULATEOVERTRAINEXTRAPOLATION($model_{curr}, D_{curr}, models_{target}$)

**Require:**
    $model_{curr}$: The size of the current model configuration.
    $D_{curr}$: The number of tokens used to train the current model.
    $model_{target}$: The size of the target model configuration.

**Ensure:**
    $m$: The calculated overtrain degree for the current configuration.
    $D_{target}$: The train token of target model under same overtrain degree.

2:     *// Part 1: Calculate overtrain degree m from the current configuration*

3:     $N_{curr} \leftarrow$ Get_N($model_{curr}$) ▷ Get $N$ (non-embedding FLOPs/token) for the current model
4:     $C \leftarrow N_{curr} \times D_{curr}$                        ▷ Calculate the total compute budget

5:     $N_{opt} \leftarrow 0.06085 \times C^{0.5445}$          ▷ Calculate Chinchilla-optimal model non-embedding FLOPs/token for budget $C$
6:     $D_{opt} \leftarrow 16.4326 \times C^{0.4555}$         ▷ Calculate Chinchilla-optimal tokens for budget $C$

7:     $\sqrt{m} \leftarrow N_{opt}/N_{curr}$                     ▷ Calculate the overtrain degree $m$
8:                                  ▷ This is equivalent to $\sqrt{m} = D_{curr}/D_{opt}$

9:     *// Part 2: Extrapolate to target model while keeping m constant*

10:     **for** each $model_t$ in $[model_{curr}] + models_{target}$ **do**
11:         $N_t \leftarrow$ Get_N($model_t$)     ▷ Get $N$ (non-embedding FLOPs/token) for the target model

12:         $N'_{opt} \leftarrow N_t \times \sqrt{m}$         ▷ Find the corresponding optimal model non-embedding FLOPs/token for the target
13:         $C_{new} \leftarrow (N'_{opt}/0.06085)^{1/0.5445}$          ▷ Derive the new compute budget

14:         $D'_{opt} \leftarrow 16.4326 \times C_{new}^{0.4555}$        ▷ Find optimal tokens for the new budget
15:         $D_{target} \leftarrow D'_{opt} \times \sqrt{m}$       ▷ Calculate the required tokens for the target model
16:     **end for**

17:     **return** $m, D_{target}$    ▷ Return the overtrain degree and the train token of target model under same $m$.
18: **end function**

---

## C   TRAINING

The model structures used in LayerMix are illustrated in Table 4. We train all the model with 2048 as the max sequence length, we use a cosine decay schedular and the initial learning rate calculated by $lr = round(0.3118 \cdot C^{-0.1250}, 8)$, the warm up ratio is set 0.5%. We use AdamW optimizer with $\beta_1 = 0.9, \beta_2 = 0.95$, weight decay= 0.1.

## D   THE RELATIONSHIP BETWEEN BENCHMARK VALIDATION LOSS AND PERFORMANCE

Our LayerMix Law focus on predicting the evaluation loss on downstream benchmarks. However, it also represents for the actual downstream performance. Figure 5 shows a near-linear relationship between validation loss and downstream performance on our evaluation tasks, and Table 5 shows the spearman corelation between validation loss and downstream performance. Lower loss consistently

Table 4: Structure of models used in LayerMix.

| Model | Hidden dim. (C) | MLP dim. (D) | Layers (L) | Heads |
|---|---|---|---|---|
| **252M** | 1024 | 2752 | 20 | 16 |
| **302M** | 1024 | 2752 | 24 | 16 |
| **392M** | 1280 | 3392 | 20 | 20 |
| **470M** | 1280 | 3392 | 24 | 20 |
| **566M** | 1536 | 4096 | 20 | 24 |
| **680M** | 1536 | 4096 | 24 | 24 |
| **850M** | 1792 | 4800 | 22 | 28 |
| **1B** | 1920 | 5120 | 24 | 30 |
| **1.2B** | 2048 | 5440 | 24 | 16 |
| **1.5B** | 2304 | 6144 | 24 | 36 |
| **1.8B** | 2304 | 6144 | 28 | 36 |
| **2.5B** | 2560 | 6848 | 32 | 40 |
| **7.7B** | 4096 | 14336 | 32 | 32 |

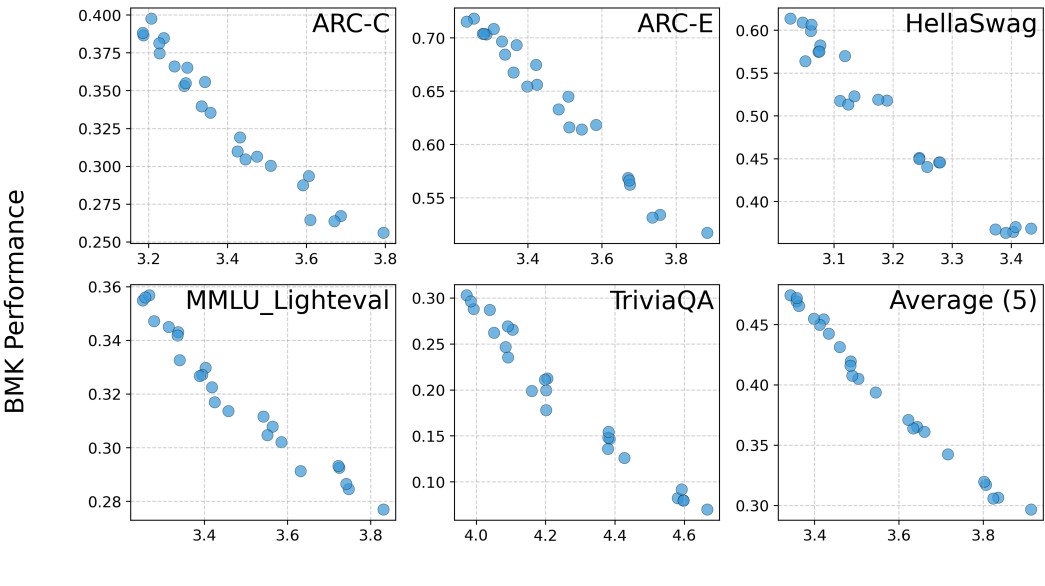

Figure 5: Validation loss versus downstream performance across benchmarks (ARC-C, ARC-E, HellaSwag, MMLU-Lighteval, TriviaQA) and their average.

corresponds to higher performance within the operating regime of our models. This indicates that improvements in loss provide reliable signals for expected gains in downstream performance.

## E    ALTERNATIVE FITS FOR $\lambda$

In Section 5.2, we model the relationship between non-embedding FLOPs/token $N$ and hyperparameter $\lambda$. Our primary specification adopts the logarithmic form Equation 9. Beyond this baseline, we also evaluated alternative function families, including an exponential form:

$$\lambda(x; a, b, c) = a \cdot \left(1 - e^{-bx+c}\right) \tag{11}$$

and a power-law form:

$$\lambda(x; a, b) = a \cdot x^b \tag{12}$$

As shown in Figure 6, the logarithmic model achieves the best fit to the $N - \lambda$ relationship, outperforming the exponential and power-law alternatives. Accordingly, we adopt function 9 as the final parameterization.

Table 5: Spearman correlation between validation loss and performance across benchmarks

| Benchmark | Spearman $r_s$ | $p$-value |
|---|---|---|
| ARC-C | -0.979 | $1.02 \times 10^{-16}$ |
| ARC-E | -0.982 | $2.72 \times 10^{-17}$ |
| HellaSwag | -0.942 | $6.13 \times 10^{-12}$ |
| MMLU-LightEval | -0.989 | $1.26 \times 10^{-19}$ |
| TriviaQA | -0.970 | $4.53 \times 10^{-15}$ |
| Average (5) | -0.996 | $3.54 \times 10^{-24}$ |

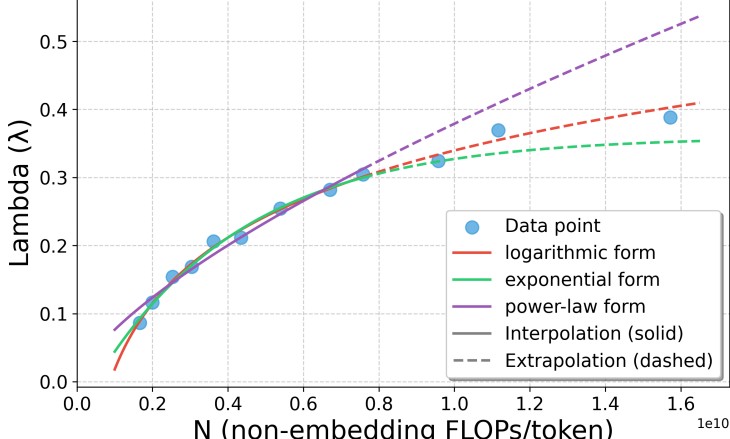

Figure 6: Comparison of functional fits for $\lambda$ as a function of $N$ (non-embedding FLOPs/token). The logarithmic form provides the best in-domain fit and extrapolation behavior compared with the exponential and power-law alternatives. Solid lines denote interpolation over observed $N$; dashed lines indicate extrapolation beyond the observed range.

## F  DEVIATION OF TRADITIONAL SCALING LAW

We show all $Loss$-$C$ curve of different LayerMix sampling weights with IST and LST in Figure 7 and Figure 8, they all exhibit a clear deviation from the traditional scaling law, which is fitted from the first three data points.

## G  QUALITY SCORE

We show some data samples in different Quality buckets in Figure 9. This figure indicates that high-score samples under our merged FineWebEdu and DCLM scores are more coherent and instructional. By contrast, low-score cases predominantly consist of advertisements or low-information content, offering little substantive value.

Table 6 reports four benchmark results for training a 1.2B model from scratch on 30B tokens using three datasets: the top 5% and top 20% selected by the FineWebEdu classifier, and a random sample,

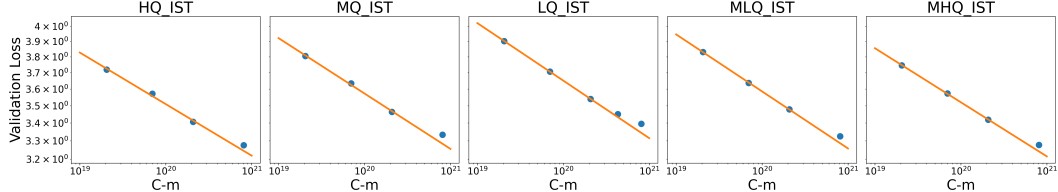

Figure 7: $Loss$ and $C_m$ Curve of different LayerMix $IST$ experiments

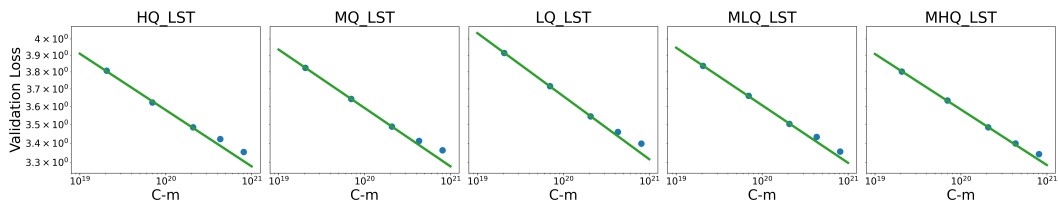

Figure 8: $Loss$ and $C_m$ Curve of different LayerMix $LST$ experiments

Table 6: FineWebEdu-selected subsets vs. random data for training a 1.2B model on 30B tokens

| Model | Data | ARC-C | HellaSwag | TriviaQA | MMLU-LightEval | avg |
|-------|------|-------|-----------|----------|----------------|-----|
| 1.2B | Random 30B | 28.50% | 51.56% | 15.55% | 30.23% | 31.46% |
| 1.2B | FWE-top20% 30B | 34.30% | 55.26% | 20.05% | 32.82% | 35.61% |
| 1.2B | FWE-top5% 30B | 37.20% | 55.14% | 19.25% | 34.50% | 36.52% |

all from Penedo et al. (2023). High-quality data selected by FineWebEdu outperforms the random baseline, and higher-quality subsets yield better results.

# H    OPTIMIZING TOKEN MIX WITH LAYERMIX LAW

We present the detailed optimal LayerMix parameters (or token-mix ratios) for different models and training budgets predicted by LayerMix Law in Table 7. This table shows that small models or small training budgets prioritize quality, while large models or large training budgets prioritize diversity.

# I    GENERALIZATION TO REFINEDWEB

To evaluate the robustness and generalization capability of the LayerMix Law across different data distributions, we conducted an additional series of verification experiments on the RefinedWeb dataset (Penedo et al., 2023).

**Experimental Setup.** We followed the identical data preprocessing, LayerMix sampling, and training procedures described in Section 3.1 and Section 5.1, with the sole exception of replacing the source corpus with RefinedWeb. Due to time and computational constraints, we limited the scope of this study to three LayerMix sampling configurations: HQ (High Quality) and LQ (Low Quality) were used for parameter fitting (interpolation), while MLQ (Medium-Low Quality) was held out for extrapolation testing. For each configuration, we trained models at three specific scales: 302M, 566M, and 1.2B parameters.

**Fitting and Extrapolation.** We applied the fitting methodology outlined in Section 5.2. Our analysis yielded two key observations:

- **Consistency of Quality Density ($f$):** The fitted values for the quality density function $f_d$ were numerically very close to those derived from our primary dataset. Specifically, the fitted parameter $\theta$ is 0.93 for RefinedWeb, which is remarkably close to the value of 0.92 obtained from our primary dataset. We attribute this similarity to the fact that RefinedWeb (Penedo et al., 2023) is also derived from Common Crawl (Common Crawl Foundation); despite employing different filtering strategies, the shared underlying data source results in a comparable information density distribution.

- **Optimization of $\lambda_N$:** In the main experiments, we modeled the relationship between the parameter $\lambda_N$ and model scale $N$ using a logarithmic curve. However, due to the limited number of data points in this verification set (only three distinct model scales), fitting a robust $\lambda_N - N$ curve was not feasible. Consequently, we skipped the curve fitting step for $\lambda_N$ and directly searched for the optimal $\lambda$ values corresponding to the specific model sizes (302M, 566M, and 1.2B).

| Quality_Range 0-5% | Quality_Range 80-100% |
|---|---|
| Celebrate your way

Whether you are having a picnic with your family, a barbeque with friends in the backyard or follow the Australian of the Year awards, Australia Day on the 26th of January is an occasion to come together as a nation and celebrate what's great about Australia.

On Australia Day we celebrate the past, present and future of the country. It is a commemoration of the day that the First Fleet landed in Sydney Cove in 1788, as well as a celebration of all the achievements of our country.

The tradition of having Australia Day as a national holiday on 26 January is actually a pretty recent one. Not until 1935 did all Australian states and territories use that name to mark that date, and only in 1994 did they begin to celebrate Australia Day as a public holiday on that date.

Today, Australia Day has grown to be a community day which is embraced by most Australians. Apart from the formal ceremonies around the country such as flag raising, citizenship ceremonies and the presentation of community awards, there are a wider range of festivities that encourage the participation of all family and community members. | This message was posted by The Dumper, posted on January 05, 2002 at 03:06:18 coming from 209.204.139 This message is a reply to Will BUY snes copier posted from Spongebob posted at January 05, 2002 at 02:25:29\n> Looking for an snes copier. Preferrably the Super Wildcard DX 2. Will pay $$$$$ or it. |
| March 7, 2012 (Shirley Allen)\nMonthly home prices in the United States increased by a seasonally adjusted 0.7 percent in December which follows a similarly revised 0.7 percent gain in November according to the Federal Housing Finance Agency's (FHFA) monthly House Price Index (HPI).\nDecember's home prices were still 0.8 percent lower than they were a year ago and since the market peak in April 2007, home prices have declined over 18 percent and are at roughly the same levels last seen in March of 2004.\n\nSix of the nine Census Divisions posted monthly price gains in December with the Mountain Division recording the most improvement of 2.5 percent. Two Divisions posted declines in home prices while one Division, the Middle Atlantic, remained unchanged from the previous month. Of the two Divisions that posted declines, the West North Central Division posted the largest decline of 0.9 percent.\nSeven of the Divisions registered year-over-year price declines with the Pacific Division posting the largest decline of 3.8 percent. The only two Divisions that posted an increase in annual home prices were the East South Central Division and the West South Central Division which posted increases of 3.0 and 1.7 percent, respectively. | these adorable little wall hangings feature bright pops of colour, tassley texture and pretty little berry knots. How can you resist?\nPlus...you get to choose from our gorge range of colours!\nMade from 100% recycled cotton and mounted on a Tasmanian Oak dowel.\nMeasures approximately 16cm wide by 33cm long (including hanger).\n*This item is hand made with love and ready to ship. Colours vary from screen to screen.\n*Looking for something similar? Custom orders are available – price available upon request.\n*Outside Australia? Please contact us for country specific freight charges.' |

Figure 9: Case study contrasting data quality. Left (0–5% quality range): coherent, informational, and instructional passages. Right (80–100% quality range): low-information, ad-like content with minimal reasoning or educational value.

Table 7: The detailed best layer token mix for different models and train token

| Model | Train Token | Source Token | $w1$ | $w2$ | $w3$ | $w4$ | $w5$ | $w6$ |
|---|---|---|---|---|---|---|---|---|
| | 200B | 500B | 0.619 | 0.376 | 0.004 | 0.001 | 0.000 | 0.000 |
| | 300B | 500B | 0.548 | 0.444 | 0.004 | 0.003 | 0.002 | 0.000 |
| | 400B | 500B | 0.496 | 0.492 | 0.007 | 0.003 | 0.002 | 0.000 |
| | 500B | 500B | 0.496 | 0.492 | 0.007 | 0.003 | 0.002 | 0.000 |
| 7B | 600B | 500B | 0.491 | 0.487 | 0.017 | 0.005 | 0.000 | 0.000 |
| | 700B | 500B | 0.439 | 0.430 | 0.130 | 0.001 | 0.000 | 0.000 |
| | 800B | 500B | 0.439 | 0.430 | 0.130 | 0.001 | 0.000 | 0.000 |
| | 900B | 500B | 0.404 | 0.403 | 0.183 | 0.006 | 0.003 | 0.000 |
| | 1000B | 500B | 0.395 | 0.387 | 0.214 | 0.003 | 0.001 | 0.000 |
| | 200B | 500B | 0.825 | 0.165 | 0.005 | 0.004 | 0.001 | 0.000 |
| | 300B | 500B | 0.619 | 0.376 | 0.004 | 0.001 | 0.000 | 0.000 |
| | 400B | 500B | 0.548 | 0.444 | 0.004 | 0.003 | 0.002 | 0.000 |
| | 500B | 500B | 0.548 | 0.444 | 0.004 | 0.003 | 0.002 | 0.000 |
| 1.8B | 600B | 500B | 0.496 | 0.492 | 0.007 | 0.003 | 0.002 | 0.000 |
| | 700B | 500B | 0.496 | 0.492 | 0.007 | 0.003 | 0.002 | 0.000 |
| | 800B | 500B | 0.496 | 0.492 | 0.007 | 0.003 | 0.002 | 0.000 |
| | 900B | 500B | 0.491 | 0.487 | 0.017 | 0.005 | 0.000 | 0.000 |
| | 1000B | 500B | 0.491 | 0.487 | 0.017 | 0.005 | 0.000 | 0.000 |
| | 200B | 500B | 0.926 | 0.066 | 0.006 | 0.002 | 0.000 | 0.000 |
| | 300B | 500B | 0.758 | 0.229 | 0.012 | 0.001 | 0.000 | 0.000 |
| | 400B | 500B | 0.619 | 0.376 | 0.004 | 0.001 | 0.000 | 0.000 |
| | 500B | 500B | 0.619 | 0.376 | 0.004 | 0.001 | 0.000 | 0.000 |
| 1.2B | 600B | 500B | 0.548 | 0.444 | 0.004 | 0.003 | 0.002 | 0.000 |
| | 700B | 500B | 0.496 | 0.492 | 0.007 | 0.003 | 0.002 | 0.000 |
| | 800B | 500B | 0.496 | 0.492 | 0.007 | 0.003 | 0.002 | 0.000 |
| | 900B | 500B | 0.496 | 0.492 | 0.007 | 0.003 | 0.002 | 0.000 |
| | 1000B | 500B | 0.496 | 0.492 | 0.007 | 0.003 | 0.002 | 0.000 |

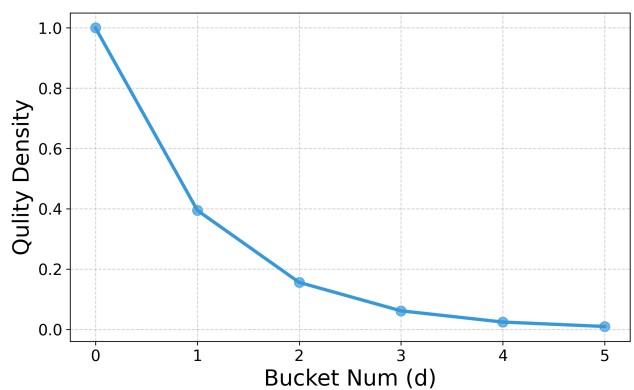

Figure 10: The fitted quality density function $f_d$ on the RefinedWeb dataset.

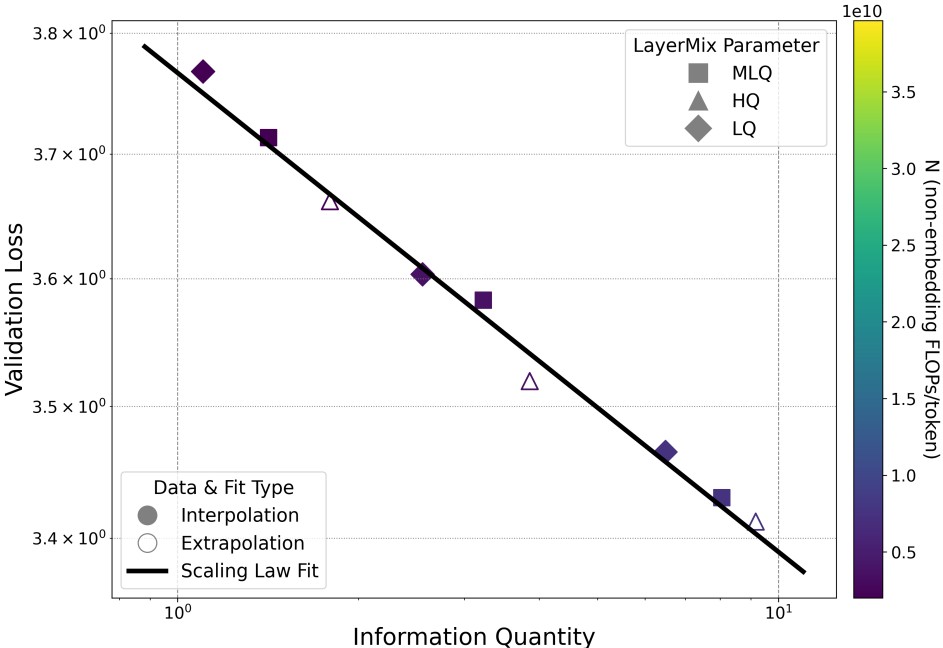

Figure 11: The Unified Information-Loss Scaling Law on the RefinedWeb dataset.

**Results.** Using the parameters fitted on the HQ and LQ configurations, we predicted the validation loss for the unseen MLQ configuration, as illustrated in Figure 11. The LayerMix Law demonstrated strong predictive accuracy on the RefinedWeb dataset, achieving a maximum absolute error of 0.36% and a mean absolute percentage Error 0.24% on the extrapolated MLQ experiments. These results further corroborate that the LayerMix Law effectively captures the fundamental trade-offs between data quality, repetition, and compute scale, independent of the specific underlying data source.

## J   LIMITATION

We note several limitations of our work. Our data bucketing is based on a fixed, empirical heuristic. We have not performed ablation studies to determine the optimal number or boundaries of these quality tiers. A more systematic approach to data partitioning could further improve the model's predictive accuracy. And while we observe that the overtrain degree $m$ systematically shifts the scaling law curve, a theoretical explanation for this behavior is still needed. These areas present clear avenues for future work.

# K    USAGE OF LLM

During the preparation of this paper, large language models (LLMs) were utilized. The use of these tools was only for polishing the language, improving grammatical structure, and performing spell checks.

