# OpenReview forum: "LayerMix Law: Scaling Law for Large Language Models on Quality-Weighted Mixture Data with Repetition"
_ICLR.cc/2026/Conference — ICLR 2026 Conference Withdrawn Submission_

### Official Review · Reviewer_3Vjy · 2025-10-20

**Soundness:** 3
**Presentation:** 3
**Contribution:** 3
**Rating:** 6
**Confidence:** 3

**Summary:**

This paper introduces LayerMix Law, a new scaling law framework designed to model large language model (LLM) performance when training data varies in quality and repetition. LayerMix Law explicitly captures the trade-off between using limited high-quality data (which may require repetition) and including lower-quality data to reduce overfitting. They propose an Information Quantity metric to quantify total learned information and demonstrate a power-law relationship between this metric and model loss. Empirically, across 27 training runs (252M–1.2B models) and extrapolations up to 7B parameters, LayerMix Law accurately predicts model loss under unseen mixtures and overtrain degrees with <1% error, providing a principled tool for data recipe optimization under data constraints.

**Strengths:**

1. Good Motivation: The paper tackles a critical and realistic problem in LLM training. It clearly explains why traditional scaling laws fail under such conditions and motivates the need for a data-aware scaling framework.
2. Comprehensive Experiments: Experiments are extensive and well-structured, covering multiple model sizes (252M–7B), data mixtures, and overtrain ratios. The consistent results across interpolation and extrapolation settings strongly support the proposed LayerMix Law.
3. Problem Formalization: The formulation of Information Quantity elegantly links data quality, repetition, and compute into a unified model. It provides a clear theoretical basis for understanding and predicting LLM performance across heterogeneous data mixtures.

**Weaknesses:**

1. Lack of Optimization Guidance: Although the law connects information gain to model loss, it does not analyze how to derive optimal mixture ratios or repetition levels under fixed compute. This limits its practical applicability for data recipe design.
2. Format and Clarity Issues:
Naming conventions like Q1_V1 and Q1_V2 are unintuitive, and equation formatting could be improved for readability.

**Questions:**

While the paper successfully establishes a theoretical connection between information gain and model performance, it stops short of providing actionable guidance on how to optimize data mixtures in practice. The LayerMix Law defines how loss depends on parameters like sampling weights $w_d$, token scale K, and repetition $R_d$, yet it does not analyze how to choose or optimize these variables to achieve the best results under a given compute budget. For example, once the relationship $L = \alpha \cdot \text{info}^{-\beta}$ is known, a natural next step would be to derive the optimal allocation of quality buckets $w_d$ that minimizes loss for fixed compute or token count. However, the paper does not explore this direction. There is no gradient-based or analytical discussion of how $w_d$, K, or S interact to yield an optimal configuration. Moreover, since the practical value of a scaling law lies in guiding future training strategies without brute-force grid search, the absence of such optimization analysis weakens the applicability of LayerMix Law for real-world data curation or scaling decisions. Including even a preliminary sensitivity or optimization study would make the framework much more actionable and impactful.

---

> ### Author Response · Authors · 2025-11-22
>
> We thank the reviewers for their careful evaluation
> ### **1. "Lack of Optimization Guidance"**
>
> **We add a section describe how to get optimal data recipe using layermix law in section 6**. Specifically, to predict the best recipe, given source data, S and training budget K, model size N, we search for the optimal token proportion w for each quality bucket. We use similar optimization process as in [1], where we initialize 100k different w, and for each w, we calculate the information quantity using LayerMix Law. Finally, we select the best recipe that optimizes the information quantity.
>
> **In table 2, we show the validation loss of the best data recipe predicted using our LayerMix Law**. It can achieve the best results among others, showing the ability of our LayerMix Law of finding the best data recipe.
>
> **We also add suggested data recipe derived by our layermix law in Table3 and also Table6 in appendix**. We observe that small models or small training budgets prioritize quality; large models or large training budgets prioritize diversity. This provide insight for optimizing training recipe without additional training cost.
>
>
>
> ### **2. "Format and Clarity Issues:"**
> **We change the name for better understanding as detailed in Table 1**
>
> From Q1 to Q5, according to the ratio of high quality buckets, we change the name to
>
> Q1：HQ (high quality),
>
> Q5: MHQ (medium-high quality),
>
> Q2：MQ (medium quality),
>
> Q4: MLQ (medium-low quality),
>
> Q3：LQ (low quality).
>
> For V1 and V2, V1 represents for infinite source token S>>K,  V2 for limited source token S=K, so we change to
>
> V1: IST (infinite source token)
>
> V2: LST (limited source token)
>
> [1] RegMix: Data Mixture as Regression for Language Model Pre-training

---

### Official Review · Reviewer_P372 · 2025-10-29

**Soundness:** 2
**Presentation:** 3
**Contribution:** 2
**Rating:** 2
**Confidence:** 5

**Summary:**

This paper introduces LayerMix Law, a data-aware scaling law for large language models (LLMs) that explicitly accounts for data quality and data repetition—two important factors often ignored by traditional scaling laws.
Existing scaling laws assume an unlimited supply of diverse, high-quality data and tend to fail when datasets are upsampled or repeated.
However, real-world LLM pretraining frequently requires upweighting scarce high-quality data or reusing existing data, leading to a fundamental trade-off between data quality and diversity.
The authors study this trade-off and propose both a theoretical formulation and empirical validation of the resulting “LayerMix Law.”
The framework extends classical scaling laws by embedding data-quality weights and repetition dynamics into performance prediction, offering a practical tool for selecting and optimizing pretraining datasets under limited compute and data availability.

**Strengths:**

1. The paper proposes a promising extension of scaling laws by incorporating data-quality weighting and repetition effects, which are crucial in today’s data-constrained LLM training.
2. The authors effectively combine ideas from scaling theory, data attribution, and mixture modeling into a single predictive law that connects information accumulation with model loss.
3. The paper reports comprehensive experiments—27 controlled pretraining runs (252M–1.2B parameters) with systematically varied data mixtures and repetition levels—and further validates extrapolation on 7B-parameter models. The resulting predictions achieve an average absolute error of only 0.15%, demonstrating impressive fit accuracy.
4. The paper is well-organized, clearly motivated, and easy to follow. Figures and explanations illustrate the intuition behind the proposed law, and the experimental design is well-documented.

**Weaknesses:**

1. The proposed information-theoretic formulation appears heuristic rather than theoretically derived. The framework would be more convincing if grounded in established information theory or accompanied by ablations comparing alternative functional forms.
2. All experiments are conducted on English Common Crawl data and relatively small models (≤7B). It remains unclear whether the fitted parameters generalize across domains (e.g., code, multilingual), which limits the broader applicability of the proposed law.
3. Although the paper cites related work (e.g., Data Mixing Laws (Ye et al., 2025), CMR Scaling Law (Gu et al., 2024)), it does not provide quantitative comparisons against these or traditional scaling-law baselines. This omission makes it difficult to assess the true improvement brought by LayerMix Law.
4. The analysis focuses exclusively on validation loss and perplexity. It would strengthen the paper to show whether the predicted improvements in pretraining loss translate into downstream task gains (e.g., MMLU, HellaSwag accuracy).

**Questions:**

See weaknesses

---

> ### Author Response · Authors · 2025-11-22
>
> We thank the reviewers for their careful evaluation
>
> ### **1. "Theoretical justification"**
>
> The concern is quite similar to Reviewer 1. We will use the same response.
>
> We have made assumption for formulation of (1)(6)(8)(9), these assumptions are made based on both intuition and empirical evidence. Specifically,
>
> **Formula 1:**  In Fisher Information theory, the accumulating speed of Fisher Information would decay to 0 when saturated. So (1) should be decreasing as t increases. And it is usually exponential decay or power-law decay.  We compared power-law vs exponential and found that exponential formula can match better.
>
> **Formula 6:** After we calculate the information quantity(IQ), we plot the L-IQ figure and found there exists clear power-law between IQ and L, i.e., single straight line in the log-log plot, as like a traditional scaling law. So we use a similar power-law function to predict L using IQ.
>
> We emphasize that the formulation of (6) reflects the rationality of design for IQ. Since the formulation of (6) highly depends on the calculation of information quantity(IQ), if the IQ itself could not reflect the influence of  repeating and quality mixing, it would be impossible to use a simple format as (6) to predict the evaluation loss. In practice, we first make sure information quantity has high correlation with L (see details in 5.2) and use (6) only to apply value mapping between IQ and L.
>
> **Formula 8:** We observe that data with higher quality yields better performance, so we design f as a decreasing function with quality bucket d. We also compared polynomial and exponential format and found that exponential works better as low quality data rarely contribute to the evaluation loss.
>
> **Formula 9:** Intuitively, $\lambda$ reflects the learning speed of the model (higher $\lambda$ will cause more decrease in info gain when the training sample repeats), so it should have positive correlation with model size since larger models will learn faster than smaller models. Then to establish the formulation, we first search for the optimal value for $\lambda$ at different model scales, and then plot the N-$\lambda$ figure (3.b) to observe the correlation. We found that the increase of $\lambda$ slows down as model size increases, and the log format best model this decay (**See comparison in Appendix E figure 6**). In figure 3.b, the dotted line well predict $\lambda$ on larger model and it proves the rational of (9)
>
> ### **2. "Experiments on different domain"**
>
> We focus on the English data of common crawl for two reasons
>
> **a. English data is more likely to have repetition issues.** According to our statistics, in CommonCrawl data there are 42% English tokens and 58% non-English tokens. However, in the training recipe, a popular large language model requires 60～80% English data. This would cause more repetition of English data. To the best of my knowledge, we are the first to systematically study the influence of quality and repetition. So we start with the most important domain.
>
> **b. To study the influence, a good definition of data quality is required.** On English data, we have fineweb-edu and dclm fasttext, which are two widely used quality filters in [1][2][3][4]. However, in other domains, **the definitions of data quality would change**, and picking a good quality filter is also an important prerequisite. So it may be quite a different problem and require lots of additional work. It is worth extending to those domains in future work. But in this paper due to the limitation of scope and time, we only focus on English data.
>
> For other English datasets, we will add experiments on different datasets to show the generalizability
>
> ### **3. "comparisons with baselines"**
>
> The mentioned two papers (data mixing law, CMR scaling law) focus on predicting mixture across domains using scaling laws. It models the effect on evaluation loss from different domains and domain intersection. In our paper, we use LayerMix Law to model how data under different quality and repeating levels affect the results, which is orthogonal to the mentioned papers.
>
> For traditional scaling laws, we show in Figure 1.a that it will fail on data with repeating. Specifically, at a larger C, where models become larger and consume tokens faster.  **We also add comparisons for ours and traditional scaling laws in the same figure for better illustration. See Figure 4.a**
>
> [1] Nemotron-CC: Transforming Common Crawl into a Refined Long-Horizon Pretraining Dataset
>
> [2] Meta-rater: A Multi-dimensional Data Selection Method for Pre-training Language Models
>
> [3] Efficient Pretraining Data Selection for Language Models via Multi-Actor Collaboration
>
> [4] QuaDMix: Quality-Diversity Balanced Data Selection for Efficient LLM Pretraining

---

> ### Author Response · Authors · 2025-11-22
>
> ### **4. “Fitting the accuracy”**
>
>    We don't directly fit the accuracy for three reasons:
>
>   1. The accuracy metric is not robust because it highly depends on the instruction following ability, which is sensitive with a small amount of sft data and change of model size, while evaluation loss has good scaling attributes. Specifically, on small models(<1B), without the SFT stage, we will usually observe random guess (25% accuracy) for MMLU throughout the training, but the MMLU loss can continue to decrease.
>
>   2. We calculated the validation loss using data from downstream tasks to improve the relationship between loss and accuracy, as described in 3.2. **We add figure and table showing high correlations between loss and accuracy within a certain range (Appendix D figure 5  table 5**) [1] also reports a high relationship between the benchmark accuracy and its evaluation loss.
>
>   3. We try to save the cost by calculating evaluation loss. Since evaluation loss only requires forward and to calculate accuracy, we need to use model generate, which is much slower than forward pass
>
> [1] Language models scale reliably with over-training and on
> downstream tasks

---

### Official Review · Reviewer_MZkc · 2025-10-29

**Soundness:** 4
**Presentation:** 4
**Contribution:** 4
**Rating:** 8
**Confidence:** 5

**Summary:**

This work studies the scaling law taking data quality and repeats into consideration. Concretely, authors split the source data into different buckets with different quality scores and then model a function of consumed tokens, model size, sampling weights, and repetition levels by  treating the training as the accumulation of information from data. And the final scaling law shows a reliable prediction across different settings.

**Strengths:**

1) It is a very important problem to fit a scaling law in terms of data quality and repeats. We are running out of data very soon. For both small and large models. And we are also looking for better compute efficiency from using more times of high quality data, but also getting struggle with the overfitting effects and diminishing return from that.
2) The insight of treating the learning as a process of information accumulation is neat and intuitively sound.

**Weaknesses:**

1) I think one assumption of this work is the definition of quality is very reliable. But how can we trust that so much?
2) Viewing the learning as a process of information accumulation is good. However, it cannot explain some aggressive overfitting -> If we repeat much more times, the model would get worse and worse when training for more steps. It is not an accumulation of info for sure.

Minor: Line 428: LayerMix Law also generate well .... "generate" -> "generalise"?
Missing reference: https://arxiv.org/abs/2305.13230 -> A very early work studying the repeat data problem also considered how the data quality matters.

**Questions:**

1) I don't understand why the name is "layer mix law". It is confusing because of the "layer" is widely used in model architecture. At the first glance, I thought this work is a model arch paper. How about "quality mix law"?
2) If we treat the learning process as a func of information accumulation, and we also take forgetting into consideration, are there any insights to order/shuffle the data for a smarter data schedule? For example, shall we repeat early more aggressively and then repeat a few more times again at the very late stage of the training to leverage the forgetting effect? No exps needed here. Just want to hear more insights.

---

> ### Author Response · Authors · 2025-11-22
>
> We thank the reviewer for the careful evaluation and recognizing the value of our work
> ### **1. "definition of quality"**
>
> We verify the reliability of quality filters by
>
> **A. Case Study.** We pick up the top and bottom cases filtered by those quality filters and observe a clear trend of more "good" cases in high quality range. **We add a case study part in Appendix G**
>
> **B. Reference.** Fineweb [1] and DCLM [2] are two popular pretrain datasets, which are generated by the Fineweb-edu and DCLM fasttext quality filters. Also, many other works on pretraining data selection choose those quality filters [3][4][5][6]
>
> **C. Empirical experiments.** We conduct experiments of pretraining a model from scratch using data selected with those quality filters, and find data with higher quality yields to better results (on aggregated benchmarks)
>
> ### **2. "Model overfitting"**
>
> That is a good point. We did observe overfitting when there are too little source tokens and too much repeating, and the results for that setting show clearly different trends from other experiments. To be honest, we do not find a neat way to even fit it as a training data point (it will cause underfitting for other data points),  not to mention predicting it. We think the aggressive overfitting setting is kind of the edge case for usual training, which is not likely to be chosen and it also shows unpredictable performance under these edge conditions, so we do not put more effort into fitting performance in these settings.
>
> ### **3. "Typo and missing reference"**
>
> **We fix typo and add discussion of the mentioned related work in the modified version**.
>
> ### **4. Name of "layer mix law"**
>
> Since we divide the data into buckets and merge data from different buckets to generate the training data, we use "layer" to indicate different buckets. That is why we call layermix. Quality mix may refer to other meanings, like merging quality scores from different quality filters.
>
> ### **5. "Training data schedule"**
>
> That is also an interesting topic related to curriculum learning. From our experience, arranging more diversified but lower quality data in the first and higher quality data but less diversified afterwards will lead to better results. Similar results have been reported in [7]
>
> [1] The FineWeb Datasets: Decanting the Web for the Finest Text Data at Scale
>
> [2] DataComp-LM: In search of the next generation of training sets for language models
>
> [3] Nemotron-CC: Transforming Common Crawl into a Refined Long-Horizon
> Pretraining Dataset
>
> [4] Meta-rater: A Multi-dimensional Data Selection Method for Pre-training
> Language Models
>
> [5] Efficient Pretraining Data Selection for Language Models via Multi-Actor
> Collaboration
>
> [6] QuaDMix: Quality-Diversity Balanced Data Selection for Efficient LLM Pretraining
>
> [7] Maximize Your Data’s Potential: Enhancing LLM
> Accuracy with Two-Phase Pretraining

---

> > ### Comment · Reviewer_MZkc · 2025-11-27
> >
> > Thanks for the reply.
> > I decide to adjust the score a little because the assumptions in this work and the rebuttal sounds a bit too strong.

---

### Official Review · Reviewer_9Rpn · 2025-10-31

**Soundness:** 3
**Presentation:** 3
**Contribution:** 3
**Rating:** 4
**Confidence:** 4

**Summary:**

The central focus of this paper is to investigate the effects of data quality and data repetition on the pre-training of large models. To this end, the authors introduce a novel metric, termed "Information Quantity," and establish its relationship with both data quality and data repetition. Subsequently, by modeling a power-law relationship between this "Information Quantity" and the val loss, the paper indirectly examines how data quality and duplication impact the model's training dynamics.

**Strengths:**

The power-law relationship established between "Information Quantity" and validation loss is robustly validated by the experiments presented in this work, demonstrating a commendable degree of originality. Furthermore, the overall experimental procedure of the paper is relatively comprehensive and complete.

**Weaknesses:**

To establish quantitative relationships, the paper hypothesizes several formulations (e.g., Equations 1, 6, 8, and 9). However, these assumptions lack sufficient justification, and their credibility is questionable.

The definitions of key concepts in the manuscript are ambiguous and appear arbitrary. Specifically, in Section 3.1 (Training Data Sampling, lines 168-174), within the formulation H(w, K, S, B), the meaning of the key variable 'S' is not introduced. Furthermore, the relationship between the associated variables 'w' and 'S' is not elucidated. The specific referent for 'B' also remains unclear. Additionally, the substitution of 'I' with f_{d} \cdot M_{d} in Equation 5, as well as the definition of f_{d} itself, requires substantial further justification or supporting evidence.

The justification for the preliminary experiments lacks rigor. Regarding the persistence of the Loss-C scaling law (line 206), demonstrating a good fit (fitting) is insufficient evidence on its own. To claim empirical validity, the authors should have used the fitted law to make further predictions; the accuracy of these predictions would then serve to validate the law. Concurrently, the "traditional law" (i.T., the OpenAI law) relates to (C-min), not simply 'C'. The authors must clarify this distinction and elaborate on the relationship between their findings and this established principle.

The Related Work section is missing several representative recent publications that are relevant to the scope of this study:

-Observational Scaling Laws and the Predictability of Langauge Model Performance

-Capability Salience Vector: Fine-grained Alignment of Loss and Capabilities for Downstream Task Scaling Law

-RegMix:Data Mixture as Regression for Language Model Pre-training

**Questions:**

1. please refer to 'Weakness';

2.How about providing some specific traindata proportion/ratio suggestions?

---

> ### Author Response · Authors · 2025-11-22
>
> We thank the reviewers for their careful evaluation of our manuscript and for the constructive comments.
>
> ### **1. "these assumptions lack sufficient justification"**
>
> These assumptions are made based on both intuition and empirical evidence. Specifically,
>
> **Formula 1:**  In Fisher Information theory, the accumulating speed of Fisher Information would decay to 0 when saturated. So (1) should be decreasing as t increases. And it is usually exponential decay or power-law decay.  We compared power-law vs exponential and found that exponential formula can match better.
>
> **Formula 6:** After we calculate the information quantity(IQ), we plot the L-IQ figure and found there exists clear power-law between IQ and L, i.e., single straight line in the log-log plot, as like a traditional scaling law. So we use a similar power-law function to predict L using IQ.
>
> We emphasize that the formulation of (6) reflects the rationality of design for IQ. Since the formulation of (6) highly depends on the calculation of information quantity(IQ), if the IQ itself could not reflect the influence of  repeating and quality mixing, it would be impossible to use a simple format as (6) to predict the evaluation loss. In practice, we first make sure information quantity has high correlation with L (see details in 5.2) and use (6) only to apply value mapping between IQ and L.
>
> **Formula 8:** We observe that data with higher quality yields better performance, so we design f as a decreasing function with quality bucket d. We also compared polynomial and exponential format and found that exponential works better as low quality data rarely contribute to the evaluation loss.
>
> **Formula 9:** Intuitively, $\lambda$ reflects the learning speed of the model (higher $\lambda$ will cause more decrease in info gain when the training sample repeats), so it should have positive correlation with model size since larger models will learn faster than smaller models. Then to establish the formulation, we first search for the optimal value for $\lambda$ at different model scales, and then plot the N-$\lambda$ figure (3.b) to observe the correlation. We found that the increase of $\lambda$ slows down as model size increases and the log formulation can best fit it (**See comparison in Appendix E figure 6**)..  In figure 3.b, the dotted line well predicts $\lambda$ on larger models and it proves the rationality of (9)
>
> ### **2. "definitions of key concepts"**
>
> Thanks for pointing it out. **We further clarify in the modified version section 3.1**
>
> S in Section 3.1 means the number of tokens in the source data, which is different to training tokens K.
>
> $w_d$ is the token proportion in d-bucket in training data. It does not correlate with S. And B, w, K and S together will affect the level of repeating.
>
> B is how we divide buckets, with $\sum B_d=1$, in this paper we use [0.05, 0.15, 0.2, 0.2, 0.2, 0.2]
>
> So in the final training data, there will be $w_d K$  tokens from bucket d, and in the source data, the tokens from bucket d is $B_d S$, yielding the repeating level of bucket d $E_d = \frac{w_d K}{B_d S}$
>
> Since we define $f_d$ as the information density in bucket d (in unit info/token), the total information in bucket d $I_d=f_d M_d$ where $M_d$ is the number of unrepeated tokens in bucket d. About the design of $f_d$, see response to question 1
>
> ### **3. "justification for the preliminary experiments lacks rigor."**
>
> In Chpt 6 EXTRAPOLATION, we valid our layermix law by predicting accuracy under different unseen conditions. Those results are summarized in figure 3.a. Specifically, we test on
>
>   **A.  Different LayerMix Sampling Weights,** as Q4,Q5 in figure 2, which verifies the performance on unseen distribution of data quality and repeating.
>
>   **B. Larger Computation**,  as Q1 Q2 Q3 in dotted line region in figure 2, which verifies the performance on larger computation with seen distribution of data quality and repeating
>
>   **C. Combination of A and B**, as Q4,Q5 in dotted line region in figure 2, which verifies the performance on larger computation with unseen distribution of data quality and repeating
>
>   **D. Different overtrain degrees**, as in figure 4, we plot the IQ-Loss line for C-m2 experiments with parameters learned from C-m1 experiments, where m2>m1 both indicate the overtrain ratio. And the IQ-Loss line for C-m2 experiments is a parallel straight line with the IQ-Loss line for C-m1 experiments, which means the parameters can extrapolate to different overtrain ratio
>
> **We also add a comparison with traditional scaling laws in figure 4.a** to show how our layermix law can predict better under repetition conditions.

---

> ### Author Response · Authors · 2025-11-22
>
> ### **4. "C-min clarification"**
>
>   **We clarify the notation in section 3.2.** In our paper we use an overtrained  setting C-m (m=3.6) as in [1] (also described in 5.1). Different from C-opt, the compute-optimal setting, C-m will consume more tokens under the same model size, which is commonly used in popular large language model training. (i.e. llama3-8B trained with 15T token) More importantly, compared with C-opt, C-m will introduce more repeating when combined with quality filtering because we train more tokens with less source tokens. That is why we conduct our studies under C-m instead of C-opt. With C-opt, the repeating level is not severe and we would not observe decay caused by repeating.
>
> ### **5. Related Work missing**
>
> We added discussion of the mentioned related work in the **section 2**.
>
> ### **6. data ratio suggestions**
>
> **We added a section in EXTRAPOLATION / Optimizing Data Recipe with LayerMix Law**, discussing how to use our layermix law to predict optimal data ratio under different training settings(model size, train size, source data size). And **we put suggested ratios in Table3 and also Table6 in appendix.** We observe that small models or small
> training budgets prioritize quality; large models or large training budgets prioritize diversity.
>
> [1] Language models scale reliably with over-training and on
> downstream tasks

---

### Note · Authors · 2026-01-27

I have read and agree with the venue's withdrawal policy on behalf of myself and my co-authors.

---

### Meta-Review · Area_Chair_Bsk1 · 2025-12-25

**Summary:**

This paper poses a new scaling law for LLMs that incorporates a notion of “information quantity”, which measures the amount of information attained from each document. The paper received widely varying reviews. Several reviewers commented on fundamental challenges with the paper: (1) insufficient justification for the proposed scaling law; (2) limited numerical analysis, especially against baselines. My own reading agrees with these concerns. The scaling law and data mixture fields are well-studied at this point with many baselines. While the paper has some comparisons with naive scaling laws, a more significant analysis with known baselines, both from the scaling law and data mixture optimization fields is necessary. At the same time, the formulation of the scaling law appears unclear and poorly motivated and needs revision for clearer exposition. In my opinion, these concerns must be addressed before the paper is ready for publication.

**Reviewer Concerns:**

- Reliance on formulations and preliminary experiments which are not sufficiently justified: The rebuttal includes some clarification, but does not address the core concern which is that the proposed framework does not naturally follow from axiomatic grounding. This may be rectified in the future by clearer exposition or revisiting the model development.
- Numerical analysis limited to small models and English CC: The paper justifies the importance of the studied setup but does not consider broader experiments.
- Limited comparison against optimal data mixing and scaling baselines: The paper and rebuttal discuss extrapolation and compare against a traditional scaling law. However, there exists a broad literature of methods in this problem area and the paper does not include sufficient comparisons against the many baselines used to analyze data mixing, data quality, and scaling.

**Reviewer Scores:**

Of the reviewers who gave negative reviews, I do not expect either to change their score. These reviewers commented on the insufficient justification and the lack of clear empirical baselines. The rebuttal did not adequately answer these questions.

---

### Decision · Program_Chairs · 2026-01-26

Reject